# The *EDN2 rs110287192* gene polymorphism is associated with paratuberculosis susceptibility in multibreed cattle population

**Mehmet Ulaş Çınar**[1,2]*, **Bilal Akyüz**[3], **Korhan Arslan**[3], **Stephen N. White**[2,4,5], **Holly L. Neibergs**[5,6], **Kadir Semih Gümüşsoy**[7]

**1** Department of Animal Science, Faculty of Agriculture, Erciyes University, Kayseri, Turkey, **2** Department of Veterinary Microbiology & Pathology, Washington State University, Pullman, WA, United States of America, **3** Department of Genetics, Faculty of Veterinary Medicine, Erciyes University, Kayseri, Turkey, **4** Animal Disease Research Unit, Agricultural Research Service, U.S. Department of Agriculture, Pullman, WA, United States of America, **5** Center for Reproductive Biology, Washington State University, Pullman, WA, United States of America, **6** Department of Animal Science, Washington State University, Pullman, WA, United States of America, **7** Department of Microbiology, Erciyes University, Faculty of Veterinary Medicine, Kayseri, Turkey

* mucinar@erciyes.edu.tr

**Citation:** Çınar MU, Akyüz B, Arslan K, White SN, Neibergs HL, Gümüşsoy KS (2020) The *EDN2 rs110287192* gene polymorphism is associated with paratuberculosis susceptibility in multibreed cattle population. PLoS ONE 15(9): e0238631. https://doi.org/10.1371/journal.pone.0238631

**Data Availability Statement:** All relevant data are within the manuscript and its Supporting Information files.

## Abstract

Paratuberculosis (pTB), also known as Johne's disease (JD), is a contagious, chronic, and granulomatous inflammatory disease of the intestines of ruminants which is caused by *Mycobacterium avium* subsp. *paratuberculosis* (MAP) infection, resulting in billions of dollars in economic losses worldwide. Since, currently, no effective cure is available for MAP infection, it is important to explore the genetic variants that affect the host MAP susceptibility. The aim of this study was to analyze a potential association between *EDN2* synonymous gene mutations (*rs110287192*, *rs109651404 and rs136707411*), that modifies susceptibility to pTB. *EDN2 rs110287192*, *rs109651404* and *rs136707411* mutations were genotyped in 68 infected and 753 healthy animals from East Anatolian Red crossbred, Anatolian Black crossbred and Holstein breed cattle by using Custom TaqMan SNP Genotyping Assays. For pTB status, serum antibody levels S/P $\geq$ 1.0 were assessed in carriers of the different *EDN2* genotypes. *EDN2 rs110287192* mutation showed a significant association with bovine pTB (adj. p < 0.05). For *rs110287192* locus, the odd ratios for GG and TG genotypes versus TT genotypes were 1.73; (95% CI = 0.34–8.59) and 0.53 (95% CI = 0.12–2.37) respectively, which indicated that proportion of TG heterozygotes were significantly higher in control animals as compared to pTB animals. On the other hand, while *rs136707411* mutation showed a suggestive association with pTB status in the examined cattle population (nominal p < 0.05); no association was detected between *rs109651404* genotypes and pTB status. Selecting animals against *rs110287192*-GG genotype may decrease the risk of pTB in cattle of the *Bos taurus taurus* subspecies.

**Funding:** This research was financially supported by the Turkish Research Council (Türkiye Bilimsel ve Teknolojik Araştirma Kurumu (TÜBİTAK)) grant number 218O128 to MUC and the funder had no role in study design, data collection and analysis, decision to publish, or preparation of the manuscript.

**Competing interests:** The authors have declared that no competing interests exist.

## Introduction

Paratuberculosis (pTB), or Johne's Disease, is a chronic disease affecting ruminant livestock, and is caused by intestinal infection with *Mycobacterium avium* subsp. *paratuberculosis* (MAP) [1]. MAP is a Gram-positive intracellular pathogen which is dependent on mycobactin, and thus unable to replicate in the environment [2]. MAP's ability to infect other animals through indirect contact is facilitated by prolonged survival times. For instance, MAP remained viable and was transmitted for up to 55 weeks in a shaded, outdoor area in Australia Whittington *et al.* [3]. Animals are usually MAP infected at a young age and are generally believed to undergo an extended latent period of chronic infection [4]. pTB begins as a localized infection that may become systemic and often results in chronic granulomatous enteritis leading eventually to weight loss, (diarrhea in some species) and death [1]. Therefore, pTB causes considerable economic losses to livestock farmers, particularly in dairy cows and beef cattle. A recent study estimated annual cost caused by pTB in the United States to be $20.80 per dairy cow [5] and this value may estimate at up to $72.5 per cow per year in Netherlands [6]. Although, data to estimate losses from pTB in beef herds are limited, Bhattarai *et al.* [7] reported an annual average loss of $276 (95% CR: $149–$478) per infected beef cattle based on survey responses. Beside direct losses as described above, indirect losses due to national and international trade restrictions and public health concerns may arise. Controversy remains as to the causation between Crohn's disease (CD) in humans and exposure to pTB, although some experiments have already shown that there is a link between pTB and CD in humans [8,9]. Despite the application of several control strategies, such as testing, vaccination and culling to reduce pTB transmission between herds, many countries continue to face challenges in controlling pTB [1]. Therefore, understanding the genetic basis of pTB susceptibility could be an alternative method for reducing the disease and selecting cattle for enhanced resistance against pTB [10,11].

Endothelins (EDN), with three isoforms of 21-residue peptides (EDN1, EDN2, EDN3), two G-protein coupled receptors (ETA and ETB), and two endothelin-converting enzymes (ECE-1 and ECE-2), are vasoconstrictor peptides [12]. The EDN are involved in the regulation of many physiological processes, such as cardiovascular development and function, craniofacial development, blood pressure regulation, renal water and sodium excretion, neurotransmission, ovulation, and proliferation, migration and differentiation of cranial, cardiac, trunk, sacral and neural crest cells [12,13]. Among EDN genes, *EDN2* has been studied in terms of ovarian research especially due to its roles in steroidogenesis and corpus luteum formation in human, model organisms and in livestock [14]. Takizawa *et al.* [15] investigated the *EDN2* expression in mice and revealed that *EDN2* mRNA was abundant in epithelial cells of the mucosal layer in the intestinal tract which may be associated with modulation of the mucosal defense by triggering immune cells. In livestock, *EDN2* has been investigated for its corpus luteum formation in cattle [16] and mRNA expression profiling in chicken tissues [17]. In addition, Settles *et al.* [18] and Neibergs *et al.* [19] reported *EDN2* as a strong functional and positional candidate gene for pTB susceptibility in Holstein cattle according to GWAS study. *EDN2* locus was identified with genome-wide significant level of association to the presence of MAP in tissue and both tissue and feces, respectively [18]. Three *EDN2* synonymous mutations on bovine chromosome 3 (BTA3), named *rs110287192*, *rs109651404* and *rs109490418* were patented for being associated with pTB susceptibility in Holstein breed cattle [20].

This study aimed to examine the association between *EDN2* SNPs *rs110287192*, *rs109651404*, *rs109490418* and pTB susceptibility in a Holstein population reared in Turkey and in Turkish indigenous cattle crossbreds (East Anatolian Red Cattle and Native Black Cattle). Genotyping experiments for three *EDN2* SNPs were conducted and the relationship with pTB susceptibility in three cattle populations was evaluated.

## Materials and methods

### Sample collection

We undertook this case-control study between June 2014 and August 2014. All experimental procedures were performed in accordance with the guidelines of the Local Ethics Committee for Animal Experiments at Erciyes University (#14/77-09.04.2014). All samples were received for confirmation of a clinical suspicion of pTB in the herd and had no further follow up. Cattle were classified as infected (cases) if they were positive for blood serum enzyme-linked immunosorbent assay (ELISA). Animals that were both clinically negative and serologically negative were considered healthy (controls). Further details regarding sample collection and ELISA diagnostic tests have been published elsewhere [21]. The study population was found to consist of 68 infected and 750 healthy animals. Briefly, blood samples were collected from cattle at two to three years of age including East Anatolian Red crossbred (n = 288), Anatolian Black crossbred (n = 112) and Holstein (n = 418) breeds from the Kayseri province and its vicinity in Turkey. Animals included in the present study were housed in similar environmental conditions and not vaccinated for pTB. Blood samples were used for the isolation of genomic DNA for genotyping, and serum samples were used for detection of MAP antibodies by ELISA.

### SNP selection

Three SNP selection methods were followed in the present study. First, we obtained genotype data of U.S. Holsteins from the existing literature on the association of *EDN2* gene polymorphisms with MAP tissue infection and pTB susceptibility [19]. Second, *rs110287192*–g.104700352T>G in 5′ UTR variant, *rs109651404*–g.104689861G>A intergenic variant and *rs109490418*–g.104706758G>A in 3′ UTR variant mutations were patented by Neibergs *et al.* patent# US20140283151 [20] for selective breeding to produce offspring having at least one of susceptibility, resistance or tolerance to pTB. Third, since assay design for the patented *rs109490418* mutation failed due to too many variants in the immediate vicinity, another mutation which is linked (D' = 0.88, $r^2$ = 0.28) and 6166 bp downstream *rs136707411*–g.104700592G>A in 5′ UTR variant region was selected for genotyping. The susceptible alleles for *rs11028192* and rs109651404 were previously reported as G and A, respectively [19].

### Genotyping

Genomic DNA was extracted from whole blood using a standard phenol–chloroform extraction procedure. DNA concentration of the samples were quantified by Nano Drop (NanoDrop, Thermo Fisher Scientific, Waltham, MA, USA), diluted to 50 ng/μl and stored at −20˚C until used. *rs110287192*, *rs109651404* and *rs136707411* SNPs of *EDN2* were genotyped using the TaqMan allelic discrimination method, which determines variants of single nucleic acid sequence. Since current SNPs have not been genotyped by using any other method, the custom TaqMan chemistry was selected as cost and time effective genotyping method. Using two primer/probe pairs in each reaction allows genotyping of the two possible variants at the single nucleotide polymorphism in a target template sequence. Details of assay IDs, primer and probe sequences were given in the Table in S1 Table. The genotyping PCR reaction was performed by adding 2 μl of genomic DNA template, 5 μl of genotyping master mix (Thermo Fisher Scientific, Waltham, MA, USA), 0.5 μl of the genotyping custom-made assay mix (probes and primers) (Thermo Fisher Scientific, Waltham, MA, USA) and 2.5 μl of DNAase-free water. Two negative controls were included on each plate. For the negative controls, 2.5 μl of DNAase-free water was added to each reaction plate instead of genomic DNA for the sample. The cycling parameters were as follows: first, denaturation was done at 95˚C for 10 min,

followed by 40 cycles of denaturation at 95˚C for 15 s, annealing and extension at 60˚C for 60 s. The PCR was performed in a StepOne Real-Time PCR System (Thermo Fisher Scientific, Waltham, MA, USA).

## Statistical analysis

An online software (http://www.husdyr.kvl.dk/htm/kc/popgen/genetik/applets/kitest.htm) was used to analyze the Hardy-Weinberg equilibrium (HWE) and allele frequency for each SNP and statistical significance was defined as $p < 0.05$. Data were analyzed using SAS 9.2 software (SAS Institute Inc., Cary, NC, USA). Additive genetic model was used for statistical analysis. The univariable analysis for logistic regression considered the infection status as a categorical response variable (yes/no), and SNPs (*all three SNPs* have three genotypes, therefore respective loci have three levels), breed (three groups i.e. two indigenous crossbred and Holstein) and sex (male and female) were included as possible explanatory variables. Genotypes were considered as ordinal variables and as class variables with the major homozygous genotype deemed as baseline. Data were analyzed using PROC LOGISTIC procedure and odds ratios (OR) with 95% confidential intervals (CIs) were calculated. Bonferroni correction (based on the total number of markers tested) was used for multiple comparisons correction, and statistical significance was defined as $p < 0.05$.

## Results

A total of 818 animals met the inclusion criteria and were included in the study to be genotyped, of which 68 had a diagnosis of pTB according to ELISA OD values ($\geq 1.0$) were subjected to association analysis and were compared to 750 age-matched healthy controls. The genotyping success rates were 97%, 94% and 90% for *rs110287192*, *rs109651404 and rs136707411*, respectively and the consensus rate (on the basis of 5% duplicates) was 100% for DNA isolated from whole blood. Although, the genotype frequencies of *rs109651404* ($\chi^2$ = 0.0042 for case and $\chi^2$ = 1.07 for control) *and rs136707411* ($\chi^2$ = 0.02 for case and $\chi^2$ = 2.98 for control*) SNPs were in accordance with the Hardy–Weinberg equilibrium in the both control group and case, genotype frequencies of *rs110287192* SNP was significantly deviated from Hardy–Weinberg equilibrium ($\chi^2$ = 35.17 for case and $\chi^2$ = 36.31 for control) due to a deficit of homozygous genotypes (TT) of the most frequent allele. The distribution of the bovine *EDN2 rs110287192*, *rs109651404* and *rs136707411* genotypes and allele frequencies in the study population are shown in Table 1.

Genotypic association analysis of all three *EDN2* polymorphisms with pTB are shown in Table 2. A significant association with the pTB was found for the *EDN2 rs110287192* variant

**Table 1. Genotype and allele distribution of the selected SNPs in animals with pTB and controls.**

| SNP | Genotypes (%) | | | Allele (%) | | $\chi^2$ ($\alpha$ = 0.05, df = 1) |
|---|---|---|---|---|---|---|
| *rs110287192* n = 796 | TT | TG | GG | T** | G | 55.31* |
| | 54 (6.78) | 463 (58.17) | 279 (35.05) | 510 (64) | 287 (36) | |
| *rs109651404* n = 769 | GG | GA | AA | G** | A | 0.97 |
| | 323 (42) | 341 (44.3) | 105 (13.7) | 500 (65) | 269 (35) | |
| *rs136707411* n = 751 | AA | GA | GG | A | G | 4.66 |
| | 291 (38.7) | 330 (43.9) | 130 (17.4) | 450 (60) | 301 (40) | |

* $p \leq 0.05$ indicates statistical significance

** Favorable allele in previous studies

**Table 2. Univariate logistic regression analysis of studied bovine *EDN2* variants and independent factors associated with pTB cases and controls.**

| SNP | Genotype | Phenotype frequency | | Nominal p-value | Adjusted p-value[a] | Fixed factors | | OR (95% CI) |
|---|---|---|---|---|---|---|---|---|
| | | Case (%) | Control (%) | | | Sex | Breed | |
| *rs110287192* | TT | 2 (3.70) | 52 (96.3) | **0.013** | | NS | * | 1.00 |
| | TG | 58 (12.53) | 405 (87.47) | | * | | | 0.53 (0.12–2.37) |
| | GG | 8 (2.87) | 271 (97.13) | | | | | 1.73 (0.34–8.59) |
| *rs109651404* | GG | 31 (9.6) | 292 (90.4) | 0.99 | | NS | ** | 1.00 |
| | GA | 30 (2.33) | 311 (91.2) | | NS | | | 0.98 (0.32–2.97) |
| | AA | 7 (6.67) | 98 (93.33) | | | | | 0.76 (0.18–3.16) |
| *rs136707411* | GG | 25 (19.23) | 105 (80.77) | **0.023** | | NS | ** | 1.00 |
| | GA | 32 (9.42) | 298 (90.58) | | NS | | | 1.66 (0.92–2.98) |
| | AA | 11 (3.78) | 280 (96.22) | | | | | 2.94 (1.34–6.46) |

Abbreviations: OR: odds ratio; 95% CI: 95% confidence interval

* p ≤ 0.05 indicates statistical significance

** p ≤ 0.01 indicates statistical significance

[a]p- value was adjusted by Bonferroni correction; NS: not significant p > 0.05

(Table 2). When the TT genotype was used as a reference, while genotype GG alone (OR = 1.73; 95% CI = 0.34–8.59; adj. p < 0.05) were significantly associated with a higher risk of pTB, genotype TG was associated with lower risk of pTB (OR = 0.53; 95% CI = 0.12–2.37; adj. p < 0.05) (Table 2). This association remained significant after Bonferroni correction for multiple tests (Bonferroni-corrected significance level for three SNPs is 0.05/3 = 0.016).

In addition, we observed suggestive association between the *EDN2 rs136707411* and increased pTB risk (nominal p = 0.023; Table 2). The association did not remain significant after Bonferroni correction for multiple tests (Table 2). At the *EDN2 rs136707411* locus, the OR of GA genotype versus GG genotype was 1.66 (95% CI = 0.92–2.98; nominal p < 0.05) and AA genotype versus GG genotype was 2.94 (95% CI = 1.34–6.46; nominal p < 0.05) which revealed that genotypes GA and AA increases the risk of pTB compared to genotype GG (Table 2). No genotype of *EDN2 rs109651404* were found to be significant associated with pTB (all p > 0.05).

## Discussion

Paratuberculosis (Johne's disease) causes a chronic diarrhea characterized by a malabsorption syndrome. The lack of absorption of nutrients in the gastrointestinal tract leads to malnutrition, muscular wasting and eventually death which results in significant economic impact worldwide [22]. Crohn's disease, a granulomatous enteritis in humans that can persist for decades, has clinical similarities with pTB in ruminants. Due to the clinical similarities between pTB and Crohn's disease, the role of MAP in Crohn's disease has been of interest [8]. Approximately 1.4 million people in North American are affected with Crohn's disease [9] and its prevalence is rapidly increasing incidence worldwide, especially in newly industrialized countries, making Crohn's as a global disease [23].

Therefore, eradication of pTB might be vital both for ruminant and public health. Control strategies to eradicate pTB mainly depend on: a) management strategies based on avoiding contact of susceptible young stock with infected animals, and b) testing animals with ELISA and culling infectious animals in herds [24]. Although management and testing strategies were powerful in reducing the infection, due to low specificity of ELISA tests and lack of effective vaccine, eradication of pTB has been shown to be difficult [24]. Thus, additional approaches,

such as genomic selection for cattle less susceptible to pTB to control pTB, are needed. Similar to our results, variability among cattle breeds in their susceptibility to pTB were identified in different experiments and support that selection for enhanced resistance to the disease is possible [21,25–27].

Association of bovine pTB susceptibility with *EDN2* was first identified with a GWAS [18] and SNP-based gene-set enrichment analysis for MAP infection detected via tissue infection or fecal shedding by using in 245 US Holsteins [19]. In a subsequent study, the *EDN2* variants *rs109651404*, *rs110287192* and *rs109490418* mutations were patented for being candidate SNPs for selection of cattle that were less susceptible to MAP infection in Holstein cows [20]. In the present experiment, *rs110287192* SNP was validated as significantly associated with pTB susceptibility in a larger cattle population that consisted of Holstein and Turkish indigenous cattle crossbreds (Table 2). For the *rs110287192* locus, the OR for TG genotypes versus TT genotypes was 0.53 (0.12–2.37; 95% CI) which revealed that the relative proportion TG genotypes was significantly higher in the control population than in the case population. It indicated that the TG genotype at the *rs110287192* locus was associated with decreased relative risk of bovine pTB and consequently selection in favor of the TG genotype or the T allele may reduce risk of pTB in cattle (Table 2). Due to the relatedness of mycobacterial pathogens such as MAP, *Mycobacterium tuberculosis* and *Mycobacterium bovis*, loci that provide less genetic susceptibility to one pathogen might afford some protection to the other organism. In fact, loci on BTA3 where we identified association for pTB susceptibility in the current study, overlapped with loci previously reported in the literature that were associated with bovine tuberculosis susceptibility [28] and bovine respiratory disease susceptibility [29].

The literature is rather sparse for identifying an association between *EDN2* variants with production or immune traits in livestock species. In cattle, pig, and laboratory animals, *EDN2* acts in the regulation of steroid production of granulosa cells [14] and *EDN2* mRNA expression found to be responsible for corpus luteum formation and ovulation [16,30,31]. Although *EDN2* was not found to be directly associated with immune traits, knockout mice for endothelin receptor B (*EDNRB*) which is a G-protein-coupled receptor of *EDN2*, developed Hirschsprung's disease (HSCR) [32]. This disease is characterized by a lack of ganglion cells of the colon and exhibits severe inflammation of the intestinal mucosa leading to like the clinical presentations associated with inflammatory bowel disease (IBD) [33]. IBD is a chronic inflammatory disease of the gastrointestinal tract in humans that can be divided into those with Crohn's disease, where disease may be present throughout the GI tract and those with Ulcerative Colitis, where disease is limited to the colon. There has been speculation that Crohn's disease may be caused by MAP as well [34].

In the present study, a strong association between a variant of *EDN2*, *rs110287192*, and pTB susceptibility in Holstein and two Turkish indigenous cattle crossbreds was demonstrated, validating, and extending the association that was previously described [18,19]. Such validation provides important support for the biological role and practical application of genomic selection for this variant [35]. Furthermore, our data also contributes to the understanding of bovine pTB and provides information that may be useful as an approach to reduce the disease through selection. Selecting against animals with the *rs110287192*-GG genotype may decrease the risk of pTB in *Bos taurus* cattle. Further analyses that are combining EDN2 genotyping and holistic expression methods through expanded sampling of other cattle breeds together with blood mRNA and serum samples for protein expression are recommended to better understand the role genomic selection could play in reducing the susceptibility to pTB in cattle.

## Supporting information

**S1 Table. Primer and probes, used for genotyping of *EDN2 rs109651404*, *rs110287192* and *rs136707411* SNPs.** F: forward; R: reverse; * assay IDs given by prob production company. (DOCX)

## Acknowledgments

The authors indebted to Ms. Codie Durfee for technical assistance during experiments.

## Author Contributions

**Conceptualization:** Mehmet Ulaş Çınar, Bilal Akyüz, Kadir Semih Gümüşsoy.

**Funding acquisition:** Mehmet Ulaş Çınar, Bilal Akyüz, Korhan Arslan, Kadir Semih Gümüşsoy.

**Methodology:** Mehmet Ulaş Çınar, Bilal Akyüz, Korhan Arslan, Kadir Semih Gümüşsoy.

**Project administration:** Mehmet Ulaş Çınar.

**Validation:** Stephen N. White, Holly L. Neibergs.

**Writing – original draft:** Mehmet Ulaş Çınar.

**Writing – review & editing:** Mehmet Ulaş Çınar, Stephen N. White, Holly L. Neibergs.

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
