## [Decision Letter · Decision Letter 0]

1 Jul 2020

PONE-D-20-15004

The EDN2 rs110287192 Gene Polymorphism is Associated with Paratuberculosis Susceptibility in Multibreed Cattle Population

PLOS ONE

Dear Dr. Çınar,

Thank you for submitting your manuscript to PLOS ONE. After careful consideration, we feel that it has merit but does not fully meet PLOS ONE’s publication criteria as it currently stands. Therefore, we invite you to submit a revised version of the manuscript that addresses the points raised during the review process.

We look forward to receiving your revised manuscript.

Kind regards,

Jasim Muhammad Uddin, DVM, PhD

Academic Editor

PLOS ONE

Additional Editor Comments:

There is a very strong link between Mycobacterium avium subspecies paratuberculosis in ruminant and Crohn's disease in humans but the cause of the later is not clear. However, have the authors found any Crohn's disease/like symptom in humans/farmers/animal handlers in the MAP-positive farm? The public health importance would be good to discuss in the manuscript.

Though this study focused on association with SNPs, the details epidemiology of the disease in cattle population should be discussed. How animals were selected? Is there any prevalence study of this disease in the region? Is there any link with age, breed, nr in farm, husbandry practice, and human?

'Funding information: This research was financially supported by the Turkish Research Council (TUBITAK) grant number 218O128.

'The funders had no role in study design, data collection and analysis, decision to publish, or preparation of the manuscript.'

4. Your ethics statement must appear in the Methods section of your manuscript. If your ethics statement is written in any section besides the Methods, please move it to the Methods section and delete it from any other section. Please also ensure that your ethics statement is included in your manuscript, as the ethics section of your online submission will not be published alongside your manuscript.

Reviewers' comments:

Reviewer's Responses to Questions

**Comments to the Author**

1. Is the manuscript technically sound, and do the data support the conclusions?

Reviewer #1: Yes

Reviewer #2: Yes

2. Has the statistical analysis been performed appropriately and rigorously? 

Reviewer #1: Yes

Reviewer #2: No

3. Have the authors made all data underlying the findings in their manuscript fully available?

Reviewer #1: Yes

Reviewer #2: Yes

4. Is the manuscript presented in an intelligible fashion and written in standard English?

Reviewer #1: Yes

Reviewer #2: Yes

5. Review Comments to the Author

Reviewer #1: Line 86-90: Do you have any data for your country (Turkey) to show how impact of this disease in your country?

Line 139: Total animal is 818 animals but in this line number you have written 68 infected and 753 healthy animals so 68+753 is not 818.

Result Part: You have 3 breeds and why not give the result for the association genotyping result individual breed. Individual association for breeding might be give strong results that SNP will affect on these 3 breeds or not. This is better to analysis SNP in each breed.

Reviewer #2: -The authors mentioned the aims of this study to analyse a potential association between EDN2 gene with pTB. To support more justification EDN2 as a candidate gene, could more explain about the position of EDN2 in chromosome is related with pTB and why the mutation that author chose synonymous mutation

- The sample were collected from three regions. How about the environment condition, are they in the same condition. Please explain more clearly

- For genotyping used custom Taqman SNP genotyping assay. It was not really clear mentioned in the methodology why used this method?

- In statistical part using proc logistic. Are this statistical model is common to used by previous study and authors were not put also the mathematical model for these analysis. Any specific reason why using Proc Logistic because normally to perform association using GLM model

- Paten number should be written in the methodology

- In Table 1, the number of animal between SNP that authors examined were different. Why the number of animal were different for example the first, second and third SNP were 796; 769 and 751 respectively

- By perform association study, is it sufficient enough to determine by choosing rs110287192-GG genotype may decrease the risk pTB in cattle without support functional study such as expression study mRNA and protein. Could authors extend to see how the expression between healthy and infected. We suggest to validate also on functional study using significant SNP rs110287192 by comparing infected and healthy animals.

6. PLOS authors have the option to publish the peer review history of their article (what does this mean?). If published, this will include your full peer review and any attached files.

Reviewer #1: No

Reviewer #2: **Yes: **Dr.agr. Asep Gunawan.Department Animal Production and Technology. Faculty of Animal Science, IPB University. Indonesia

---

## [Author Response · Author response to Decision Letter 0]

23 Jul 2020

ONE-D-20-15004

The EDN2 rs110287192 Gene Polymorphism is Associated with Paratuberculosis Susceptibility in Multibreed Cattle Population

PLOS ONE

Dear Dr. Çınar,

Thank you for submitting your manuscript to PLOS ONE. After careful consideration, we feel that it has merit but does not fully meet PLOS ONE’s publication criteria as it currently stands. Therefore, we invite you to submit a revised version of the manuscript that addresses the points raised during the review process.

We look forward to receiving your revised manuscript.

Kind regards,

Jasim Muhammad Uddin, DVM, PhD

Academic Editor

PLOS ONE

Comment: Additional Editor Comments:

Comment 1: There is a very strong link between Mycobacterium avium subspecies paratuberculosis in ruminant and Crohn's disease in humans but the cause of the later is not clear. However, have the authors found any Crohn's disease/like symptom in humans/farmers/animal handlers in the MAP-positive farm? 

Response 1: There are strong clues that Mycobacterium avium subspecies paratuberculosis (MAP) the cause of a chronic intestinal disease in domestic and wild ruminants may cause of idiopathic inflammatory bowel disease (IIBD). Crohn’s disease is one of the IIBD form together with ulcerative colitis. However, in the present investigation, we did not follow up neither consumer’s health status nor farmers. To the best of our knowledge in Turkey, there is no study that investigate the prevalence of Crohn’s disease that caused by MAP. Nevertheless, in literature, experiments, were performed in other countries showed the MAP zoonotic transmission from domestic animals to humans. 

Wynne JW, Bull TJ, Seemann T, et al. Exploring the zoonotic potential of Mycobacterium avium subspecies paratuberculosis through comparative genomics. PLoS One. 2011;6:e22171.

Pierce, E. S. (2018). Could Mycobacterium avium subspecies paratuberculosis cause Crohn’s disease, ulcerative colitis… and colorectal cancer? Infectious Agents and Cancer, 13(1), 1-6.

In the manuscript, there is a sentence in the discussion “IBD is a chronic inflammatory disease of the gastrointestinal tract in humans that can be divided into those with Crohn’s disease, where disease may be present throughout the GI tract and those with Ulcerative Colitis, where disease is limited to the colon. There has been speculation that Crohn’s disease may be caused by MAP as well [34].”

34 Pierce, E. S. (2018). Could Mycobacterium avium subspecies paratuberculosis cause Crohn’s disease, ulcerative colitis… and colorectal cancer? Infectious Agents and Cancer, 13(1), 1-6.

Comment 2: The public health importance would be good to discuss in the manuscript.

Though this study focused on association with SNPs, the details epidemiology of the disease in cattle population should be discussed. How animals were selected? Is there any prevalence study of this disease in the region? Is there any link with age, breed, nr in farm, husbandry practice, and human?

Response 2: Animals were selected according to clinical observations. Cattle, showed clinical signs of pTB as cachexia and diarrhea, were subjected to experiment. All, animals were in the same age; they were in 2-3 years old and exposed to the same environmental condition. The same region previously investigated by Gumussoy et al. (2015). There pTB prevalence was found 12.2% for Holstein and in our experiment, prevalence was identified as 11.7% which was cited in our previous article Cinar et al. 2018 and that was cited in the current manuscript as [21]. 

Gümüşsoy, K. S., Ica, T., Abay, S., Aydin, F., & Hizlisoy, H. (2015). Serological and molecular diagnosis of paratuberculosis in dairy cattle. Turkish Journal of Veterinary and Animal Sciences, 39(2), 147-153.

Cinar MU, Hizlisoy H, Akyüz B, Arslan K, Aksel EG, Gümüşsoy KS. Polymorphisms in toll-like receptor (TLR) 1, 4, 9 and SLC11A1 genes and their association with paratuberculosis susceptibility in Holstein and indigenous crossbred cattle in Turkey. J Genet. 2018;97: 1147–1154.

Comment 1: 1. Please ensure that your manuscript meets PLOS ONE's style requirements, including those for file naming. The PLOS ONE style templates can be found at

Response 1: Thanks for reminder, we have checked and modified accordingly.

Comment 2: 2. Thank you for stating the following in the Acknowledgments Section of your manuscript:

'Funding information: This research was financially supported by the Turkish Research Council (TUBITAK) grant number 218O128.

'The funders had no role in study design, data collection and analysis, decision to publish, or preparation of the manuscript.'

a. Please clarify the sources of funding (financial or material support) for your study. List the grants or organizations that supported your study, including funding received from your institution.

d. If you did not receive any funding for this study, please state: “The authors received no specific funding for this work.”

Response 2: We have removed funding information from the acknowledgement and stated in the online Funding Statement section.

Comment 3: Please include your amended statements within your cover letter; we will change the online submission form on your behalf.

Response 3: We have given our amendment in the cover letter.

Comment 4: 3. Please include captions for your Supporting Information files at the end of your manuscript, and update any in-text citations to match accordingly. Please see our Supporting Information guidelines for more information: http://journals.plos.org/plosone/s/supporting-information

Response 4: Revised accordingly.

Comment 5: 4. Your ethics statement must appear in the Methods section of your manuscript. If your ethics statement is written in any section besides the Methods, please move it to the Methods section and delete it from any other section. Please also ensure that your ethics statement is included in your manuscript, as the ethics section of your online submission will not be published alongside your manuscript.

Response 5: Thanks for reminder. I have deleted the “Compliance with ethical standards:” section in the revised version of manuscript. 

Review Comments to the Author

Reviewer #1

Comment 1: Reviewer #1: Line 86-90: Do you have any data for your country (Turkey) to show how impact of this disease in your country?

Response 1: There are few literatures about the local (province level) prevalence of paratuberculosis in Turkey, done by using different detection methods such as serology and PCR. But those experiments have subjected limited number of animals and farms. Additionally, no comprehensive sampling and analyzing method was applied in national level, therefore, no economic analysis had been generated in which deals with the loss which is caused by paratuberculosis in Turkey. 

Comment 2: Line 139: Total animal is 818 animals but in this line number you have written 68 infected and 753 healthy animals so 68+753 is not 818.

Response 2: Sorry for inconvenience. There was a typing error. The number of healthy animals were corrected as 750 in the revised manuscript.

Comment 3: Result Part: You have 3 breeds and why not give the result for the association genotyping result individual breed. Individual association for breeding might be give strong results that SNP will affect on these 3 breeds or not. This is better to analysis SNP in each breed.

Response 3: Thank for the comment. There were two reasons not to apply single breed analysis instead of multi-breed analysis. First, the number of cases were lower in native-crossbreds compared to Holsteins. There were 7 pTB cases and 4 pTB cases were detected in East Anatolian Red crosbred and Anatolian Black crosbred, respectively. On the other hand, 57 pTB cases were identified in the Holsteins. Hence, we thought, since our aim was to identify association of the mutations with the pTB status, we used multi-breed approach. Second, researcher stated the advantages of using multiple breedsfor association analysis in literature and applied this method in many animal species (Ramayo-Caldas et al. 2016, Bianchi et al. 2015, van den Berg, Boichard, Lund, 2016).Karlsson et al. (2007) stated that using multiple breeds that share the same phenotype, may decrease the noise, and help to identify selected loci accurately.

Karlsson, E., Baranowska, I., Wade, C. et al. Efficient mapping of mendelian traits in dogs through genome-wide association. Nat Genet 39, 1321–1328 (2007).

Ramayo-Caldas, Y., Renand, G., Ballester, M., Saintilan, R., & Rocha, D. (2016). Multi-breed and multi-trait co-association analysis of meat tenderness and other meat quality traits in three French beef cattle breeds. Genetics Selection Evolution, 48(1), 37.

Bianchi, M., Dahlgren, S., Massey, J., Dietschi, E., Kierczak, M., Lund-Ziener, M., & Ollier, W. E. (2015). A multi-breed genome-wide association analysis for canine hypothyroidism identifies a shared major risk locus on CFA12. PLoS One, 10(8), e0134720.

van den Berg, I., Boichard, D., & Lund, M. S. (2016). Comparing power and precision of within-breed and multibreed genome-wide association studies of production traits using whole-genome sequence data for 5 French and Danish dairy cattle breeds. Journal of Dairy Science, 99(11), 8932-8945.

Reviewer #2

Comment 1: The authors mentioned the aims of this study to analyse a potential association between EDN2 gene with pTB. To support more justification EDN2 as a candidate gene, could more explain about the position of EDN2 in chromosome is related with pTB and why the mutation that author chose synonymous mutation

Response 1: In livestock, EDN2 has been investigated for its corpus luteum formation in cattle. However, no research was devoted for the association of bovine EDN2 with immune traits. EDN2 mutations, selected for the current work taken from the previous studies. Settles et al. [18] and Neibergs et al. [19] reported EDN2 as a strong functional and positional candidate gene for pTB susceptibility in Holstein cattle according to GWA study. Three EDN2 synonymous mutations, named rs110287192, rs109651404 and rs109490418, were patented for being associated with pTB susceptibility in Holstein breed cattle [20]. 

18. Settles M, Zanella R, McKay SD, Schnabel RD, Taylor JF, Whitlock R, et al. A whole genome association analysis identifies loci associated with Mycobacterium avium subsp. paratuberculosis infection status in US holstein cattle. Anim Genet. 2009;40: 655–662. doi:10.1111/j.1365-2052.2009.01896.x

19. Neibergs HL, Settles ML, Whitlock RH, Taylor JF. GSEA-SNP identifies genes associated with Johne’s disease in cattle. Mamm Genome. 2010;21: 419–425. doi:10.1007/s00335-010-9278-2

20. Neibergs HL, Ricardo Z, Taylor JF, Wang Z, Scraggs E, White SN, et al. Compositions and methods for diagnosis of genetic susceptibility, resistance, or tolerance to infection by mycobacteria and bovine paratuberculosis using promoter variants of EDN2. Google patents [Preprint]. 2014 [cited 2020 April 17]. Available: https://patents.google.com/patent/US20140283151

Response 2: Although EDN2 has been investigated for its corpus luteum formation in cattle

Comment 2: - The sample were collected from three regions. How about the environment condition, are they in the same condition. Please explain more clearly

Response 2: All animals were subjected to same environmental condition. Samples were taken in summer and they were kept in the same farm that they were fed with the same ratio. “Animals included in the present study were housed in similar environmental conditions” statement has already been mentioned in the sample collection section. 

Comment 2: - For genotyping used custom Taqman SNP genotyping assay. It was not really clear mentioned in the methodology why used this method?

Response 2: The mutations, were genotyped in the present work derived from a previous GWAS study. TaqMan genotyping method could be one of the most cost and time efficient method for these kinds of mutations identified by GWAS. TaqMan system is one of the earliest methods for SNP genotyping, based on fluorescently-tagged, allele-specific probes detected using real-time polymerase chain reaction (PCR)-based assays. Since mutations did not genotyped before elsewhere, no literature mentioned about RFLP enzyme for DNA digestion. Therefore, we had to designed custom TaqMan assays for the respective mutations. 

The sentence was added in the revised manuscript: “Since current SNPs have not been genotyped by using any other method, the custom TaqMan chemistry was selected as cost and time effective genotyping method.”

Osaki, R., Imaeda, H., Ban, H., Aomatsu, T., Bamba, S., Tsujikawa, T., & Andoh, A. (2011). Accuracy of genotyping using the TaqMan PCR assay for single nucleotide polymorphisms responsible for thiopurine sensitivity in Japanese patients with inflammatory bowel disease. Experimental and therapeutic medicine, 2(5), 783-786.

Comment 3: - In statistical part using proc logistic. Are this statistical model is common to used by previous study and authors were not put also the mathematical model for these analysis. Any specific reason why using Proc Logistic because normally to perform association using GLM model

Response 3: Binary logistic regression is useful where the dependent variable is dichotomous (e.g., healthy/sick, live/die, graduate/dropout, vote for A or B). Proc Logistic is very similar to Proc GLM, although it has a binary outcome variable rather than an interval outcome.If the outcome is ordinal, Proc Logistic can also be used, but with a complementary log-log link function instead of the more standard log function. Logistic regression also defines odds ratios for the input variables. There are similar articles analyze the same type of data by using Proc Logistic. 

Vázquez P, Ruiz-Larrañaga O, Garrido JM, et al. Genetic association analysis of paratuberculosis forms in holstein-friesian cattle. Vet Med Int. 2014;2014:321327. doi:10.1155/2014/321327

Ruiz-Larrañaga O, Garrido JM, Manzano C, et al. Identification of single nucleotide polymorphisms in the bovine solute carrier family 11 member 1 (SLC11A1) gene and their association with infection by Mycobacterium avium subspecies paratuberculosis. J Dairy Sci. 2010;93(4):1713-1721. doi:10.3168/jds.2009-2438

Bhaladhare A, Sharma D, Kumar A, et al. Single nucleotide polymorphisms in toll-like receptor genes and case-control association studies with bovine tuberculosis. Vet World. 2016;9(5):458-464. doi:10.14202/vetworld.2016.458-464

Mohd Baqir, Saket Bhushan, Amit Kumar, Arvind Sonawane, Ranvir Singh, Anuj Chauhan, Ramji Yadav, Om Prakash, Renjith R, Aashish Baladhare & Deepak Sharma (2016) Association of polymorphisms in SLC11A1 gene with bovine tuberculosis trait among Indian cattle, Journal of Applied Animal Research, 44:1, 380-383, doi: 10.1080/09712119.2015.1091333 

Comment 4: - Paten number should be written in the methodology

Response 4: The sentence was modified in the revised manuscript.

From:intergenic variant and rs109490418–g.104706758G>A in 3′ UTR variant mutations were patented by [20]

To:intergenic variant and rs109490418–g.104706758G>A in 3′ UTR variant mutations were patented by Neibergs et al. patent# US20140283151 [20]

Comment 5: - In Table 1, the number of animal between SNP that authors examined were different. Why the number of animal were different for example the first, second and third SNP were 796; 769 and 751 respectively

Response 5: Thanks for this point. The reason why animal number showing variability was the missing genotypes. Due to limited budget, we had no chance to repeat animals. However, animal numbers per genotype is still over 200, which is recommended for reliable association analysis.

Comment 6: - By perform association study, is it sufficient to determine by choosing rs110287192-GG genotype may decrease the risk pTB in cattle without support functional study such as expression study mRNA and protein. Could authors extend to see how the expression between healthy and infected. We suggest validating also on functional study using significant SNP rs110287192 by comparing infected and healthy animals.

Response 6: The current experiment was aimed to identify the favorable genotype for pTB resistancy. Reviewer is right to ask whether EDN2 was the correct candidate gene for pTB susceptibility. Authors are planning for to continue the functional genomic analysis such as RNA-Seq and Western Blot analysis for EDN2 gene. However, this needs animal should be bought from farmers after validated to be pTB. Further financial applications are warranted to perform functional genomics experiments in order to investigate pTB susceptibility.

---

## [Decision Letter · Decision Letter 1]

18 Aug 2020

PONE-D-20-15004R1

The EDN2 rs110287192 Gene Polymorphism is Associated with Paratuberculosis Susceptibility in Multibreed Cattle Population

PLOS ONE

Dear Dr. Çınar,

Thank you for submitting your manuscript to PLOS ONE. After careful consideration, we feel that it has merit but one of the reviewer has additional queries. Therefore, we invite you to submit a revised version of the manuscript that addresses the points raised during the review process.

We look forward to receiving your revised manuscript.

Kind regards,

Jasim Muhammad Uddin, DVM, PhD

Academic Editor

PLOS ONE

Additional Editor Comments (if provided):

No more comments from me but please address the queries from Dr Gunawan.

Reviewers' comments:

Reviewer's Responses to Questions

**Comments to the Author**

1. If the authors have adequately addressed your comments raised in a previous round of review and you feel that this manuscript is now acceptable for publication, you may indicate that here to bypass the “Comments to the Author” section, enter your conflict of interest statement in the “Confidential to Editor” section, and submit your "Accept" recommendation.

Reviewer #1: All comments have been addressed

Reviewer #2: All comments have been addressed

2. Is the manuscript technically sound, and do the data support the conclusions?

Reviewer #1: Yes

Reviewer #2: Partly

3. Has the statistical analysis been performed appropriately and rigorously? 

Reviewer #1: Yes

Reviewer #2: Yes

4. Have the authors made all data underlying the findings in their manuscript fully available?

Reviewer #1: Yes

Reviewer #2: Yes

5. Is the manuscript presented in an intelligible fashion and written in standard English?

Reviewer #1: Yes

Reviewer #2: Yes

6. Review Comments to the Author

Reviewer #1: The author was response from the reviewers clearly and corrected according to the suggestion as well. The manuscript is currently in scientific standard for publication in this journal. However, the next scientific report in deeply still needed. Hope the author using the scientific to solve this problem in livestock production.

Reviewer #2: The authors have revised their manuscript and taken some, but not all of my earlier comments into account.

1. It is still unclear what the the EDN2 position or is located in the BTA (Bos Taurus) chromosom. Please meantion it in which chromosome?. If the authors mentioned are taken from GWAS study explain litle bit information in the manuscript to support these gene is important related to pTB.

2. Authors have explained that the animal which used in this study were in the same environment condition. The sample were collected in summer time based on reviewed response. Could authors added in the manuscript including also the age of cattle to emphesized that only effect of genetic EDN2 that would be obsreved.

3. In line 224-225, authors observed suggestive association between the EDN2

rs136707411 and increased pTB risk. Its better to put the level of p-value which is categorized as suggestive association forisntance (P<0.10) etc

4. Line 251-252: No genotype of EDN2 rs109651404 were found to be significant

252 associated with pTB (all p > 0.05). This sentence could be merge with previous paragraph because only consist few sentences.

5 Line 308-311, Further analyses through expanded sampling of other cattle breeds are recommended to better understand the role genomic selection could play in reducing the susceptibility to pTB in cattle. Based on previous comment please add not only the large number of cattle which should be added, but also it is needed functional study of EDN2 at the level of mRNA and protein expression. Authors could seen paper related below:

Gunawan, A., K. Kaewmala, M.J. Uddin, M. U. Cinar, D. Tesfaye, C. Phatsara, E. Tholen, C. Looft, and K. Schellander. 2011. Association study and expression analysis of porcine ESR1 as a candidate gene for boar fertility and sperm quality. Anim. Reprod. Sci. 128: 11-21.

Gunawan, A., K. Kaewmala, M.J. Uddin, M. U. Cinar, D. Tesfaye, C. Phatsara, E. Tholen, C. Looft, and K. Schellander. 2011. Investigation on association and expression of ESR2 as a candidate gene for boar sperm quality and fertility. Domest. Reprod. Anim. doi:10.1111/j.1439-0531.2011.01968.x

7. PLOS authors have the option to publish the peer review history of their article (what does this mean?). If published, this will include your full peer review and any attached files.

Reviewer #1: No

Reviewer #2: **Yes: **Prof. Dr.agr. Asep Gunawan, SPt, MSc

---

## [Author Response · Author response to Decision Letter 1]

19 Aug 2020

PONE-D-20-15004R1

The EDN2 rs110287192 Gene Polymorphism is Associated with Paratuberculosis Susceptibility in Multibreed Cattle Population

Additional Editor Comments (if provided):

Comment 1: No more comments from me but please address the queries from Dr Gunawan.

Response 1: Thanks editor for improving the quality of manuscript.

Reviewer Comments:

Reviewer #1: 

Comment 1: The author was response from the reviewers clearly and corrected according to the suggestion as well. The manuscript is currently in scientific standard for publication in this journal. However, the next scientific report in deeply still needed. Hope the author using the scientific to solve this problem in livestock production.

Response 1: Thanks reviewer for improving the quality of manuscript. We are continuing to work in pTB, we are going to inform Ministry of Agriculture in Turkey in terms of our results. They may evaluate results and may suggest to use in selection programs.

Reviewer #2: 

Comment 1: The authors have revised their manuscript and taken some, but not all of my earlier comments into account.

Response 1: We tried to answer all points. Sorry for inconvenience, we answer all points in this version.

Comment 2: 1. It is still unclear what the the EDN2 position or is located in the BTA (Bos Taurus) chromosom. Please meantion it in which chromosome?. If the authors mentioned are taken from GWAS study explain litle bit information in the manuscript to support these gene is important related to pTB.

Response 2: Chromosome location of three EDN2 mutations were given and details of previous GWAS study was mentioned in the manuscript. The sentence was modified as:

In addition, Settles et al. [18] and Neibergs et al. [19] reported EDN2 as a strong functional and positional candidate gene for pTB susceptibility in Holstein cattle according to GWAS study. EDN2 locus on bovine chromosome 3 was identified with genome-wide significant level of association to the presence of MAP in tissue and both tissue and feces respectively [18]. Three EDN2 synonymous mutations on bovine chromosome 3 (BTA3), named rs110287192, rs109651404 and rs109490418 were patented for being associated with pTB susceptibility in Holstein breed cattle [20]. 

Comment 3: 2. Authors have explained that the animal which used in this study were in the same environment condition. The sample were collected in summer time based on reviewed response. Could authors added in the manuscript including also the age of cattle to emphesized that only effect of genetic EDN2 that would be obsreved.

Response 3: Age of cattle was mentioned in the sample collection sub-section in line 132 in the revised manuscript.

Comment 4: 3. In line 224-225, authors observed suggestive association between the EDN2

rs136707411 and increased pTB risk. Its better to put the level of p-value which is categorized as suggestive association forisntance (P<0.10) etc

Response 4: p value was added in the revised version (L228).

Comment 5: 4. Line 251-252: No genotype of EDN2 rs109651404 were found to be significant

252 associated with pTB (all p > 0.05). This sentence could be merge with previous paragraph because only consist few sentences.

Response 5: The sentence was merged with the previous paragraph as asuggested.

Comment 6: 5 Line 308-311, Further analyses through expanded sampling of other cattle breeds are recommended to better understand the role genomic selection could play in reducing the susceptibility to pTB in cattle. Based on previous comment please add not only the large number of cattle which should be added, but also it is needed functional study of EDN2 at the level of mRNA and protein expression. Authors could seen paper related below:

Gunawan, A., K. Kaewmala, M.J. Uddin, M. U. Cinar, D. Tesfaye, C. Phatsara, E. Tholen, C. Looft, and K. Schellander. 2011. Association study and expression analysis of porcine ESR1 as a candidate gene for boar fertility and sperm quality. Anim. Reprod. Sci. 128: 11-21.

Gunawan, A., K. Kaewmala, M.J. Uddin, M. U. Cinar, D. Tesfaye, C. Phatsara, E. Tholen, C. Looft, and K. Schellander. 2011. Investigation on association and expression of ESR2 as a candidate gene for boar sperm quality and fertility. Domest. Reprod. Anim. doi:10.1111/j.1439-0531.2011.01968.x

Response 6: There is no experimental infection unit in Turkey suitable for bovine experiments. However, we are planning to collect slaughterhouse tissue and organ samples from pTB positive and negative animals too. Thanks for sample articles and suggestion. We have revised the sentence accordingly.

Further analyses that are combining genotyping and holistic gene expression methods through expanded sampling of other cattle breeds together with blood mRNA and serum protein samples are recommended to better understand the role genomic selection could play in reducing the susceptibility to pTB in cattle.

---

## [Editor Report · Decision Letter 2]

21 Aug 2020

The EDN2 rs110287192 Gene Polymorphism is Associated with Paratuberculosis Susceptibility in Multibreed Cattle Population

PONE-D-20-15004R2

Dear Prof Cinar

We’re pleased to inform you that your manuscript has been judged scientifically suitable for publication and will be formally accepted for publication once it meets all outstanding technical requirements.

Kind regards,

Jasim Muhammad Uddin, DVM, PhD

Academic Editor

PLOS ONE

Additional Editor Comments (optional):

No further comments. I am pleased to inform you that the manuscript is acceptable.
---

## [Editor Report · Acceptance letter]

25 Aug 2020

PONE-D-20-15004R2 

The *EDN2 rs110287192* gene polymorphism is associated with paratuberculosis susceptibility in multibreed cattle population 

Dear Dr. Çınar:

I'm pleased to inform you that your manuscript has been deemed suitable for publication in PLOS ONE. Congratulations! Your manuscript is now with our production department. 

Kind regards, 

on behalf of

Dr. Jasim Muhammad Uddin 

Academic Editor

PLOS ONE